# Weakly Supervised Data Augmentation Through Prompting for Dialogue Understanding

**Maximillian Chen**[1]*, **Alexandros Papangelis**[2], **Chenyang Tao**[2], **Andy Rosenbaum**[2],
**Seokhwan Kim**[2], **Yang Liu**[2], **Zhou Yu**[1] **Dilek Hakkani-Tur**[2],
[1]Columbia University, [2]Amazon Alexa AI
maxchen@cs.columbia.edu, zy2461@columbia.edu
{papangea, chenyt, andros, seokhwk, yangliud, hakkanit}@amazon.com

## Abstract

Dialogue understanding tasks often necessitate abundant annotated data to achieve good performance and that presents challenges in low-resource settings. To alleviate this barrier, we explore few-shot data augmentation for dialogue understanding by prompting large pre-trained language models and present a novel approach that iterates on augmentation quality by applying weakly-supervised filters. We evaluate our methods on the emotion and act classification tasks in DAILYDIALOG and the intent classification task in FACEBOOK MULTILINGUAL TASK-ORIENTED DIALOGUE. Models fine-tuned on our augmented data mixed with few-shot ground truth data are able to approach or surpass existing full-shot state-of-the-art performance on both datasets. For DAILYDIALOG specifically, using 10% of the ground truth data we outperform the current state-of-the-art model which uses 100% of the data.

## 1 Introduction & Related Work

Most common ways of automatic data augmentation in natural language tasks include simple perturbations [Wei and Zou, 2019, Karimi et al., 2021, Xie et al., 2020] and generative approaches [Kim et al., 2021, Sahu et al., 2022, Edunov et al., 2018]. However, these methods do not utilize intersentential context, which is essential to encode for both dialogue understanding and generation.

On the other hand, modern pre-trained language models (PLMs) can be prompted to complete dialogues using prefix prompts [Liu et al., 2021], which naturally encode conversational context. PLMs also have shown impressive zero- and few-shot capabilities [Brown et al., 2020, Bommasani et al., 2021] in dialogue tasks and have been successfully used in generative augmentation frameworks for tasks such as intent classification [Sahu et al., 2022, Li et al., 2021], commonsense reasoning [Yang et al., 2020], and response generation [Kulhánek et al., 2021, Gao et al., 2020b]. Several studies examine in-context learning, which involves including training examples as part of a prompt [Wei et al., 2022, Min et al., 2022, Chen et al., 2022, Lu et al., 2022]. In this work, we take the first step towards applying few-shot prompting to augmenting dialogue datasets. We focus on low-resource[1] settings, contributing an empirical account of augmenting turn-level dialogue understanding tasks using discrete prompting which encodes dialogue history as in-context examples.

One challenge with zero- and few-shot prompting with PLMs is that the outputs may exhibit more diversity than one would expect for a specific task, which confounds model training [Perez et al., 2021, Zhao et al., 2021]. Specifically, PLMs often synthesize data points which lie outside of the data

---

*Work done during internship at Amazon Alexa AI
[1]both in terms of data and cost of computational resources.

(neutral) Alice: You're going to set up your own law office, aren't you?
(neutral) Bob: Yes. After so many years of hard work, I'd rather I had an office of my own.
(happy) Alice: If you need help, don't hesitate to ask me.
(happy) Bob: I'll be very glad if you would help.
(happy) Alice: I'd like to wish you every success in your new venture.
(happy) Bob: Thank you. I wish I would.
(happy) Alice:
**Generated Responses:**
1. Good luck to you. Let's do lunch soon, Bob.
2. It's such a rare pleasure to meet such an ideal partner in your work.
3. You know, you seem quite different.

Figure 1: Example augmented conversation from DAILYDIALOG with a generated turn following the desired emotion "happy." WEAKDAP filters out generated turns which do not follow the label (red).

manifold[2] of a given task, instead following the distribution of the generic pretraining corpora. Due to their distance from the target task's distribution, these augmented samples may be considered low quality. We thus propose WEAKDAP (Weakly supervised Data Augmentation through Prompting), a framework that iteratively improves the quality of augmented data in dialogue classification tasks by introducing a weakly supervised labeler to filter prospective data points. Figure 1 demonstrates WEAKDAP filtering out a low-quality synthetic utterance. We demonstrate the effectiveness of WEAKDAP on emotion and dialogue act classification in DAILYDIALOG [Li et al., 2017], showing on-par or better performance compared to state-of-the-art full-shot results by augmenting only 10% of the original data. We additionally examine the robustness of WEAKDAP using a separate task: cross-lingual augmentation for Spanish intent detection in FBTOD [Schuster et al., 2019].

## 2 Data Augmentation Methods

Our approach consists of two parts: prompting PLMs using dialogue context, and applying weak supervision to refine prompt-augmented datasets.

### 2.1 Constructing Dialogue Prompts

Dialogue contexts can be used to form prefix prompts which serve as the input to a PLM[3]. We augment the data by replacing dialogue turns, which are selected using the dialogue context construction strategies below. We illustrate specific examples of each in Figure 2 and Section E in the Appendix. Each generated utterance can be prescribed a randomly sampled or ground truth reference label.

**Conversation Trajectory Augmentation (`CTA`).** We take each speaker's first turn as ground-truth context and iteratively replace the next turn with a generated utterance. We autoregressively use each generated utterance as context to generate the next turn. Each ground truth conversation results in one synthetic conversation with a new "trajectory."
**All-Turn Augmentation (`ATA`).** ATA iteratively replaces each turn in the conversation with a generated utterance, but uses the ground truth context instead of the generated context. For a conversation with $n$ turns, this results in $n-1$ "new" conversations of length 2 through $n$.
**Last-Turn Augmentation (`LTA`).** This is a special case of ATA where we simply choose the last turn of the conversation to replace with a generated utterance. This results in the largest conversational context, helping guide the conditional output closer to the ground truth language manifold. Relative to a ground-truth conversation, this yields one new conversation, with an alternate last turn. Example in Figure 1.

---

[2]Kim et al. [2021] hypothesizes that synthetic data must lie along the same natural language manifold as the ground truth data, proposing linear interpolation among existing data.

[3]While augmentation by prompting PLMs can help expand linguistic diversity, it can also introduce biases which exist in PLMs' pre-training corpora. Additionally, it may underline biases in the existing low-resource data being augmented. We discuss this further in Appendix A.

> **Emotion Augmentation with GPT-J (Original Emotion)**
> Alice in a neutral mood: Oh you look awful! What's the matter?
> Bob in a neutral mood: Oh! I feel really under the weather. I've got a sore throat and a bad cough.
> Alice in a neutral mood: Oh dear. Maybe you've caught a cold.
> Bob in a neutral mood: Yes, I've had lots of overtime to do recently and I haven't slept much at all.
> Alice in a neutral mood: Well then, you should get some rest this weekend and don't go out drinking.
> Bob in a neutral mood:
>
> **Result:**
> Thanks, but I can't afford to do that.

> **Emotion Augmentation with GPT-J (Swapped Emotion)**
> Alice in a neutral mood: Oh you look awful! What's the matter?
> Bob in a neutral mood: Oh! I feel really under the weather. I've got a sore throat and a bad cough.
> Alice in a neutral mood: Oh dear. Maybe you've caught a cold.
> Bob in a neutral mood: Yes, I've had lots of overtime to do recently and I haven't slept much at all.
> Alice in a neutral mood: Well then, you should get some rest this weekend and don't go out drinking.
> Bob in a surprised mood:
>
> **Result:**
> What's that supposed to mean?

Figure 2: Example conversation augmentation prompt for emotion classification using GPT-J, prescribing the original ground-truth emotion (left) and a randomly sampled emotion (right). This is augmented using Last Turn Augmentation, i.e., the first five turns are taken from the ground-truth data and the model is asked to generate the sixth and final turn. Both boxes represent a new augmented conversation when taken in aggregate.

## 2.2 Augmentation with Weak Supervision

While prompting large PLMs provides a convenient, powerful way to bridge the gap between inadequate training data and data-hungry conversational models, there is a caveat: those PLMs are trained on generic corpora (*i.e.*, web crawls, books, *etc.*), whose distribution may considerably differ from the data needed to train task-specific models (*e.g.*, see Figure 4). This motivates *post-hoc* adjustments to make our prompted augmentations more task-aware. Weak supervision has been proposed for finding a "useful representation" for a task [Robinson et al., 2020]. Intuitively, naive prompted augmentations are less potent because they lack task-knowledge[4], which can be distilled from ground-truth ("gold") samples by training an auxiliary model. We can then use that model to filter out inconsistent generated utterances.

We propose WEAKDAP, a framework generalizeable to any prompt-based augmentation task. In this work, we prompt GPT-J 6B [Wang and Komatsuzaki, 2021] and the Alexa Teacher Model (ATM) 20B [Soltan et al., 2022]. As Figure 3 illustrates, WEAKDAP consists of three parts. We first augment the "gold" data and train a task classifier on the gold and "silver" data. Then, we iteratively re-augment the data and re-train the classifier. For the augmentation step on each iteration, we use the classifier trained during the previous iteration to create a weak silver label for each generated instance, and filter out instances where the silver label does not match the prescribed label with high confidence, i.e., low entropy. We reason that data points which a weak labeler thinks are labeled incorrectly with low confidence could still be useful for learning during training (further discussion in Section G in the Appendix). Moreover, this indicates that their labels may be in fact be correct. To this end, we filter out incorrect instances classified in the bottom 80th percentile of entropy, computed as in the equation below, where C is the number of classes and $p_i$ is the probability of class $i$.[5]

$$Entropy = \sum_{i}^{C} -p_i * log_2(p_i)$$

This weakly guarantees that the generated data is not of low-quality. This continues until the classifier's performance doesn't improve by at least $\epsilon$ for $k$ rounds. Here, we fix $\epsilon = 0.005, k = 3$.

---

[4]PLMs only see prompts during generation; to fully account for task knowledge one should include all available examples in-context, which is generally impractical.

[5]This threshold is tunable.

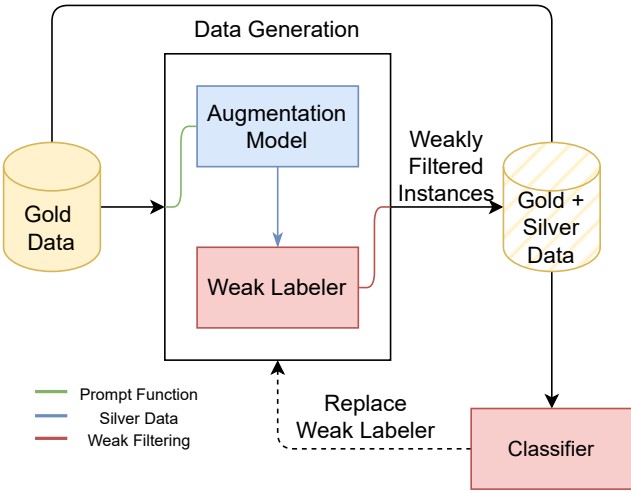

Figure 3: The workflow of WEAKDAP. On each iteration, the Gold Data is augmented by replacing conversation turns generated by providing a PLM with prefix prompts. Each prospective silver training instance is weakly classified as either following its intended label or not, using a task specific classifier. The gold and silver data are used as training data for the next generation's classifier. This process repeats until the performance of the classifier does not improve past a threshold.

**Other Task-Aware Augmentation Approaches.**     Similar task-aware generative augmentation approaches typically distill task-knowledge into the generator. Yang et al. [2020] proposes augmentation for commonsense reasoning by fine-tuning two generators (for answering and distracting) and relabelling synthetic data points using a task model, while Papangelis et al. [2021] fine-tunes a generator using reinforcement learning. With large PLMs, these methods are costly and less practical. While few-shot prompting is a cheaper solution, it is less effective at encoding lots of task knowledge, as in-context example capacity is limited. WEAKDAP bridges the gap between prompt-based augmentation with little task-knowledge and complex mechanisms with higher computational costs; it does not need to fine-tune the generator, as we prompt it using dialogue context as in-context utterance examples.

## 3   Experiments

We benchmark various augmentation methods on the classification tasks in DAILYDIALOG, a high-quality open-domain dialogue dataset, and the intent detection task of FBTOD, a task-oriented dialogue dataset (dataset details in Figure C).

### 3.1   DAILYDIALOG Emotion Classification

We first conduct a thorough evaluation of our augmentation methods using the emotion classification task in DAILYDIALOG as a case study, in the full and few-shot settings[6]. For our augmentation model, we use GPT-J 6B[7] [Wang and Komatsuzaki, 2021], which is one of the largest causal language models publicly available, and has been able to achieve performance competitive to GPT-3 on many tasks [Wang, 2021, Black et al., 2022]. For all DAILYDIALOG experiments we use the Speaker Turn Model (STM) [He et al., 2021], a RoBERTa [Liu et al., 2019]-based classification model with speaker turn awareness[8], as the classification task model and weak labeler.

There are seven emotion labels: *neutral, anger, disgust, fear, happiness, sadness*, and *surprise*. Each label is a rich, descriptive token on its own, so in constructing a prompt, we directly use it as an adjective (e.g., "Alice in a *happy* mood:"). Additionally, we conjecture that directly using conversation history forms the best set of in-context examples to generate utterances which convey

---

[6]We randomly sample 1%, 5%, and 10% of the data.

[7]We examined OPT-30B [Zhang et al., 2022], but it was far slower without large performance improvements.

[8]STM achieves state-of-the-art performance on full-shot DAILYDIALOG act classification (87.5% accuracy).

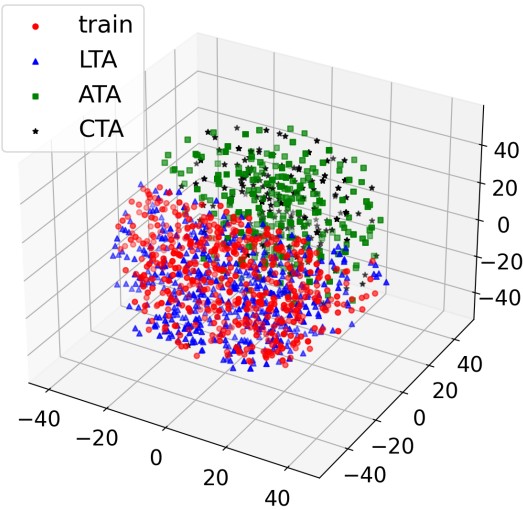

Figure 4: t-SNE projection of a random sample (for clarity) of training and augmented data.

a prescribed emotion while remaining within the gold data manifold. Example prompts provided in Figure 2 (`LTA`) and Section E in the Appendix (`ATA`, `CTA`). In Figure 4, we compare the data manifold of the synthetic data resulting from `LTA` and `CTA` with a maximum augmentation size of 2x that of the amount of original data used, and `ATA` which results in a size between 5.7x and 6.2x (see Appendix Table 3). We can see that `CTA` results in a separate cluster of data, likely due to the underlying distribution of GPT-J's pretraining data. `ATA`'s distribution has a clear distinct cluster as with `CTA`, but also posseses some overlap with the training distribution. In contrast, we see that synthetic data from `LTA` lies within the training data manifold. This is likely due to the in-distribution context provided as conditional input to GPT-J. We hypothesize that this context most closely guides synthetic data towards the original data distribution. This may be more beneficial in tasks where we do not expect distribution shift from training to inference. In tasks such as response generation, where diverse output is desirable, it may be more beneficial to have training data that falls into an expanded manifold as with `CTA` or `ATA`. Comparing the three methods, we find that `CTA` and `ATA` (Appendix Figure 7, Table 3) are beneficial, but underperform `LTA` (Figure 5) for this task. Thus, we primarily experiment using `LTA`.

In low-resource settings, it is important to quantify how much ground truth data is available. When augmenting seed data, it is essential to quantify how much data is being generated (i.e., by providing a Size Multiplier relative to the amount of seed data used; Feng et al. [2020]). Figure 5 shows the performance of STM in each data setting with varying amounts of synthetic data resulting from `LTA`. Following existing work, we report micro F1 ignoring the majority label, due to a heavy imbalance towards the neutral emotion [Liang et al., 2021, Lee and Choi, 2021, Wang et al., 2020]. We see that STM with augmented 10% data reaches an F1 score of 0.686 and 0.70 with augmented full-shot data, surpassing the existing state-of-the-art of 0.641 set by S+PAGE [Liang et al., 2021]. We observe that adding too much augmented data eventually hurts performance in each setting, further suggesting that prompting alone has inconsistent quality. 2x is the best performing multiplier and we use that to conduct all other experiments. Figure 6 also shows that WEAKDAP improved STM's performance in each few-shot setting. On average, the labeler reduced the augmented data size from 2x to 1.8x, indicating that it may foster more efficient learning.

Prior work has not found large differences in performance between backtranslation and other perturbation methods despite higher computational overhead [Xie et al., 2020, Kim et al., 2021]. Thus, we primarily compare against EDA [Wei and Zou, 2019][9] and AEDA [Karimi et al., 2021][10] as noise injection baselines. As seen in Figure 6, using noise perturbed data underperforms normal data, an outcome corroborated by Kumar et al. [2020], Chen et al. [2021]. Finally, we examined the importance of context by experimenting with a standard in-context learning prompt with 10 randomly

---

[9]EDA includes synonym replacement, random insertion, random swap, and random deletion.

[10]AEDA randomly inserts punctuation marks into text.

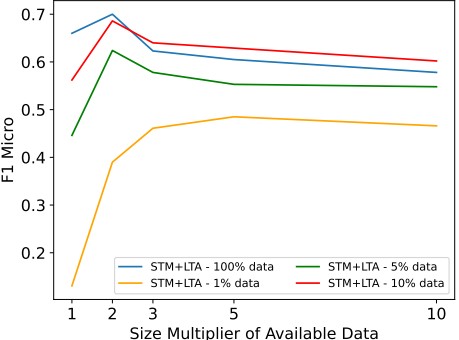
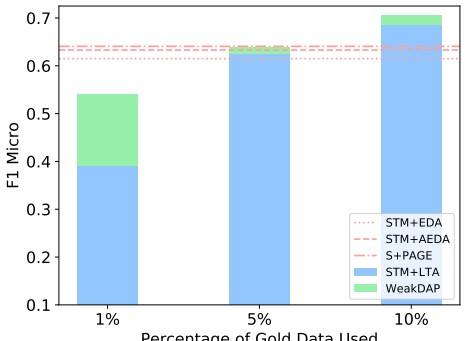

Figure 5: Micro F1 on the DAILYDIALOG emotion classification task using LTA for different data sizes.

Figure 6: Comparison of prompt-augmentation with LTA and WEAKDAP. WEAKDAP with 10% data outperforms full-shot state-of-the-art performance with S+PAGE, and baseline augmentation approaches mixed with full-shot data.

sampled utterances of the same emotion label [Sahu et al., 2022, Brown et al., 2020], finding that dialogue context consistently outperforms random sampling (e.g. 0.624 versus 0.600 F1 using 5% data).

## 3.2 DAILYDIALOG Act Classification

There are four dialogue act labels: *inform, question, directive*, and *commissive*. While these individual terms are less descriptive than emotion labels, they form descriptive tokens if used as active verbs (e.g., "Alice directs Bob:"). See Section E in the Appendix for an example.

Following our findings on Emotion Classification, we augment the few-shot DAILYDIALOG Act Classification task with LTA while providing full-shot performance for reference. Our results in Table 1 indicate that WEAKDAP results in the highest accuracy across all few-shot settings, although using STM on the unaugmented data achieves the highest F1 in the 5% setting. Noticeably, the performance improvements on the act classification are not as drastic as in the emotion classification task. This is likely because the baseline performance on the act classification task was already competitive.

## 3.3 Cross-lingual Augmentation

To assess the robustness of WEAKDAP, we additionally evaluated its performance on a completely different setting: low-resource cross-lingual augmentation for intent classification on FBTOD. For augmentation, we used the Alexa Teacher Model [Soltan et al., 2022], a 20B multi-lingual seq2seq language model pre-trained on a text de-noising objective similar to Liu et al. [2020] using large publicly available datasets including mC4 [Xue et al., 2021] and Wikipedia. For intent classification, we fine-tuned XLMRoBERTa (XLMR; Conneau et al. [2020]).

Full-shot fine-tuning of XLMR yields an accuracy of $98.8\%$ on the Spanish test set, however we focus on the few-shot Spanish setting, with few- and full-shot cross-lingual augmentation from English. For all few-shot datasets, we use $1\%, 5\%,$ and $10\%$ of the original training data for that language (see Section F). FBTOD is a single-turn dataset, so we adapt the in-context learning prompt in Sahu et al. [2022], mixing the reference Spanish instance with randomly sampled English instances of the same label for in-context learning (example in Section E in the Appendix). We do not examine Thai because the model has not been pre-trained on Thai data.

We augment the few-shot Spanish setting using both low-resource and high-resource English data. With low-resource English, we use the same percentage of data as the ground-truth Spanish data.

| DAILYDIALOG | Accuracy | F1 |
| --- | --- | --- |
| 1% No Aug. | 0.789 | 0.648 |
| 1% Prompt | 0.792 | 0.714 |
| + WeakDAP | **0.796** | **0.716** |
| 5% No Aug. | 0.817 | **0.809** |
| 5% Prompt | 0.828 | 0.762 |
| + WeakDAP | **0.832** | 0.765 |
| 10% No Aug. | 0.839 | 0.802 |
| 10% Prompt | 0.839 | 0.815 |
| + WeakDAP | **0.842** | **0.820** |
| 100% SotA | 0.875 | — |

Table 1: Accuracy and Micro F1 (ignoring majority label) on DailyDialog Act classification using STM with few-shot data. We augment using LTA resulting in data sizes of at most two times the original data size.

| FBTOD ES | $\mathrm{Acc}_{LR}$ | $\mathrm{F1}_{LR}$ | $\mathrm{Acc}_{HR}$ | $\mathrm{F1}_{HR}$ |
| --- | --- | --- | --- | --- |
| 1% No Aug. | 0.572 | 0.164 | 0.572 | 0.164 |
| 1% Prompt | 0.737 | 0.316 | 0.776 | 0.359 |
| + WeakDAP | **0.834** | **0.495** | **0.831** | **0.528** |
| 5% No Aug. | 0.845 | 0.417 | 0.845 | 0.417 |
| 5% Prompt | 0.953 | 0.641 | 0.954 | 0.682 |
| + WeakDAP | **0.957** | **0.715** | **0.962** | **0.732** |
| 10% No Aug. | 0.942 | 0.588 | 0.942 | 0.588 |
| 10% Prompt | 0.973 | 0.772 | 0.973 | 0.791 |
| + WeakDAP | **0.979** | **0.905** | **0.976** | **0.846** |
| 100% No Aug. | 0.988 | 0.889 | 0.988 | 0.889 |

Table 2: Accuracy and Macro F1 on the FBTOD Spanish dataset. $HR/LR$: High-resource (full-shot) and Low-resource (few-shot matching the Spanish percentage) English Training data.

Table 2 indicates that fine-tuning XLMR on WEAKDAP outperforms prompt-based augmentation in all settings both in terms of accuracy and F1 score.

## 4 Conclusion

We contribute significant progress towards few-shot prompt-based augmentation for dialogue tasks. We introduce WEAKDAP and demonstrate its augmentation quality filtering capabilities by surpassing full-shot state-of-the-art performance with few-shot examples on DAILYDIALOG and achieving strong few-shot performance on FBTOD. In the future, we will examine ways to integrate soft prompting into WEAKDAP as well as identify an appropriate feedback mechanism for response generation.

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

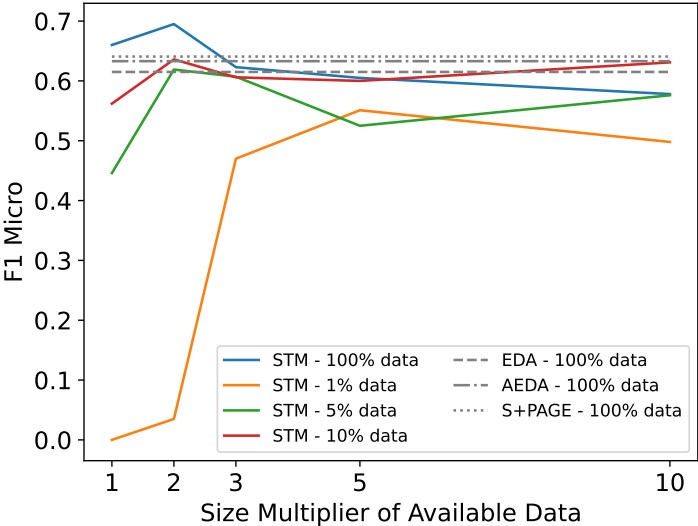

Figure 7: Classification results on the DAILYDIALOG emotion classification task using Conversation Trajectory Augmentation. Multiplier size represents the dataset size in multiples of the amount of gold data. $n\%$ represents the percentage of the gold data used.

## A   Ethical Considerations

**Pre-trained Language Model Biases**   In this work, we directly use only two datasets: DAILY-DIALOG and FBTOD. However, large pre-trained language models like GPT-J have already been pre-trained on massive corpora such as The Pile [Gao et al., 2020a], which is a webcrawl of much of the internet. While this promotes diverse language, this also provides no guarantee over the types of content that the model is capable of producing. It is possible that the model could generate negative, hurtful, or offensive content [Gao et al., 2020a, Christian, 2020]. Prompting methods as the ones proposed in this work fundamentally rely on large pre-trained language models. As such, it is possible that this hurtful content could leak into augmented datasets, if left unchecked. While in this work we only discuss dialogue understanding tasks, this may present a bigger issue in dialogue generation where hurtful utterances may actually be relayed to users. There are several works that attempt to mitigate these issues, for example Lauscher et al. [2021], Schick et al. [2021], Barikeri et al. [2021], Faal et al. [2022], Pavlopoulos et al. [2020].

**Data Biases**   In a similar vein, every dataset, including DAILYDIALOG and FBTOD has its own biases. While introducing language model output from another language manifold (i.e., a pre-trained language model's training corpora) will help to lessen some of the biases present in unaugmented datasets, generation conditioned solely on existing dialogue contexts may continue to reinforce some of these existing biases.

**Reproducibility**   Upon acceptance, we plan to release all of the augmentation code, as well as the seed data (i.e. each few-shot setting) used in all experiments.

## B   Limitations

In this work, we consider two different dialogue contexts — social chit-chat in DAILYDIALOG and multi-lingual task-oriented dialogue in FACEBOOK MULTILINGUAL TASK-ORIENTED DIALOGUE. Due to computational constraints, it is difficult to consider several dialogue contexts with all of our experimental settings. However, we believe that we present a set of experiments representative of the scope of our method's generalizability to different dialogue understanding tasks and integratability with different pre-trained language models.

Figure 8: Example prompt using Conversation Trajectory Augmentation for emotion classification using GPT-J. The original conversation (left) has four turns. After allowing each speaker to speak once, GPT-J autoregressively generates the rest of the conversation. Turn 3 (top right) is generated using the first two ground truth turns, and Turn 4 (bottom right) is generated using the first two ground truth turns and the generated third turn. The final candidate conversation is represented by all of the turns in the bottom right box.

**Prompt Selection** Much recent research focuses on how to design prompts to optimize performance on a variety of different tasks Liu et al. [2021]. In this work, we did not apply such prompt engineering methods. Instead, we focused on the conversational nature of prompting for augmentation in dialogue tasks. Our prompt primarily consists of dialogue context. Design decisions included the use of the names "Alice" and "Bob," as well as the choice to encode instance labels in natural language form. We did not formally evaluate these decisions, but we subjectively saw that the use of other names can yield similar performance. We also noticed that using names generally provided better results than generic speaker tags such as "Speaker 1" and "Speaker 2."

## C Datasets

DAILYDIALOG [Li et al., 2017] is a high-quality open-domain conversation dataset. The official DAILYDIALOG training set contains 11,118 dialogues, while the validation and test sets each have 1,000 dialogues. Each conversation is annotated with a topic label and has on average eight turns. Additionally, each turn is annotated with a dialogue act and an emotion label.

Facebook Multilingual Task-Oriented Dialogue (FBTOD; Schuster et al. [2019]) is a dataset which contains single-turn task-oriented utterances in English, Spanish, and Thai. Each utterance is annotated with one of twelve intent labels.

## D DAILYDIALOG Experiments

In order to measure and isolate the effect of prompting methods, we hold many of the experimental settings fixed. For data generation with GPT-J, we use top-$p$ sampling with $p = 0.92$. The resulting generated data is parsed from the decoded language model outputs. All of our downstream classification experiments are performed using STM. In order to isolate the effects of augmentation and due to computational limitations, we fix the tunable STM parameters with an initial learning rate of .0001, 2

**Original Conversation**
Alice in a surprised mood: My goodness! She is thirty seconds faster than the world record in 5000 metres race.

Bob in a surprised mood: Excuse me, what did you say?

Alice in a happy mood: A Chinese girl has broken the world record in the Olympic Games.

Bob in a surprised mood:  That's incredible. I can't believe it.

Alice in a happy mood: You have to. It is sure.

Bob in a surprised mood: So, what's her name?

**Augmented Conversation 1**
Alice in a surprised mood: My goodness! She is thirty seconds faster than the world record in 5000 metres race.
Bob in a surprised mood: Excuse me, what did you say?
Alice in a happy mood:

**Augmented Conversation 2**
Alice in a surprised mood: My goodness! She is thirty seconds faster than the world record in 5000 metres race.
Bob in a surprised mood: Excuse me, what did you say?
Alice in a happy mood: A Chinese girl has broken the world record in the Olympic Games.

**Generated Turn 4:**
I am dying to know who is the girl?

**Augmented Conversation 3**
Alice in a surprised mood: My goodness! She is thirty seconds faster than the world record in 5000 metres race.
Bob in a surprised mood: Excuse me, what did you say?
Alice in a happy mood: I said my goodness, that girl is fast.
Bob in a surprised mood: That's incredible. I can't believe it.
Alice in a happy mood:

**Generated Turn 5:**
This is the first time in the history of the world.

**Augmented Conversation 4**
Alice in a surprised mood: My goodness! She is thirty seconds faster than the world record in 5000 metres race.
Bob in a surprised mood: Excuse me, what did you say?
Alice in a happy mood: I said my goodness, that girl is fast.
Bob in a surprised mood: That's incredible. I can't believe it.
Alice in a happy mood: You have to. It is sure.
Bob in a surprised mood:

**Generated Turn 6:**
You mean that it is a Chinese girl that has broken the world record?

Figure 9: Example prompt using All Turn Augmentation for emotion classification using GPT-J. The original conversation (left) has four turns. After allowing each speaker to speak once, GPT-J generates turn three using the first two ground truth turns (top right), turn four using the first three ground truth turns (middle right), turn five using the first four ground truth turns (bottom left), and turn six using the first five ground truth turns (bottom right). Each of the blue boxes represents a new conversation with the corresponding generated turn as the new endpoint.

recurrent layers and a $50\%$ dropout layer. We let STM fine-tune for up to 100 epochs; we use early stopping with a patience of 10 epochs.

All experiments are implemented with `PyTorch` [Paszke et al., 2019] and HuggingFace's `Transformers` [Wolf et al., 2020], and run on AWS *p3.16xlarge* EC2 instances.

## D.1 Conversation Trajectory Augmentation

In Figure 7, we see that we are able to reach state-of-the-art performance on the emotion classification task using conversation trajectory augmentation as well. Augmented full-shot STM reaches an F1 score of 0.695. In all except the $1\%$ ground-truth setting, Conversation Trajectory Augmentation underperforms Last Turn Augmentation. This is likely due to the fact that Conversation Trajectory Augmentation departs from the ground-truth data distribution, in contrast to Last Turn Augmentation as displayed in Figure 4. This is not necessarily a downside of generating new trajectories - it just indicates that it does not perform as strongly on this particular classification task. By definition, training a model on data that closely matches the testing distribution will be advantageous during evaluation. However, beyond classification tasks, it is possible that this diverse data distribution will be more favorable, e.g., in response and dialogue generation, where one would want to see more diverse responses.

## D.2 All-Turn Augmentation

As displayed in Table 3, All-Turn Augmentation generally does not outperform either Conversation Trajectory Augmentation nor Last-Turn Augmentation. Moreover, we see that the resulting augmented sizes are roughly six times that of the original data. The closest comparable augmented size used

**Act Augmentation with GPT-J**
Alice informs Bob: Good morning.
Bob informs Alice: Er, good morning, yes, er...
Alice directs Bob: I'm phoning about the job that was in the paper last night.
Bob directs Alice: Oh, yes. Erm, well, could you tell me your name, please?
Alice informs Bob: Oh, Candida Fawcett.
Bob questions Alice: Oh, yes. Erm, well, what exactly is it that interests you about the job?
Alice informs Bob: Well, I just thought that it was right up my street you know.
Bob directs Alice: Really, hmmmm. Erm, well, could you perhaps tell me a little about yourself?
Alice informs Bob: Yes, arm, I'm 23. I've been working abroad, I'm um...
Bob informs Alice:

**Result:**
Well, perhaps you would like to think it over. Erm, and er...

Figure 10: Example prompt using Last Turn Augmentation for act augmentation using GPT-J. The original conversation had nine turns, and the augmented conversation is the same except with the last turn replaced with a generated utterance. GPT-J learns to use some of the speaker's tendencies, such as using the term "Erm."

| Size | Augmented Mult. | F1 |
|------|-----------------|-------|
| 1%   | 5.7x            | 0.459 |
| 5%   | 6.2x            | 0.463 |
| 10%  | 6.1x            | 0.610 |

Table 3: Resulting augmentation sizes and classification F1 Micro using All-Turn Augmentation on the DAILYDIALOG emotion classification task.

with Last-Turn and Conversation Trajectory Augmentation is a multiplier of five, but at that multiple, either Last Turn or Conversation Trajectory Augmentation yields the strongest performance for each data setting.

# E  Example Prompts

We present several examples of prompts corresponding to the different methods of conversation augmentation.

Figure 2 shows an example of Last Turn Augmentation and demonstrates that by omitting the last turn of the original conversation, we can form a prefix prompt that allows GPT-J to generate an utterance according to a prescribed emotion, whether it be the ground truth emotion from the training data or a user-defined emotion. The ground truth turns given as context combined with the generated utterance constitute a new augmented conversation.

Figure 8 shows an example of Conversation Trajectory Augmentation. In order to set the context for each speaker, we always include the first ground truth turn for each speaker in the context. For each subsequent turn, we autoregressively let the generation model generate the utterance by feeding in the previously generated utterance. This process continues until the number of turns in the new augmented conversation reaches the length of the original ground truth conversation.

We show an example of All Turn Augmentation in Figure 9. We again include the first ground truth turn for each speaker in the conversation. However, instead of autoregressively feeding in previously generated utterances, the context of the generated turn $i$ is always the ground truth turns 1 through $i - 1$. Since the generated turn $i$ is not guaranteed to form a coherent conversation when with ground truth turns $i + 1$ through $n$ (where $n$ is the length of the ground truth conversation), we consider each generated turn to be an endpoint for a new conversation.

```
[CLM] The following sentences belong to the same
category weather/find:
Example 1 (in English): tell me the weather report for
half moon bay
Example 2 (in English): is it going to snow today
Example 3 (in English): What is the current weather
Example 4 (in English): What is the KATU weather
report?
Example 5 (in English): is it going to rain this week?
Example 6 (in Spanish): Cuál es la humedad hoy?
Example 7 (in Spanish):

Result:
¿cuál es el pronóstico del tiempo?
```

Figure 11: Cross-lingual prompt between Spanish and English for FBTOD intent detection. [CLM] is a token reserved for the model that is required for in-context prompts.

Finally, we show an example using Last Turn Augmentation while prescribing a DAILYDIALOG act label in Figure 10. We see that GPT-J correctly generates an utterance that follows the act "inform," and is even able to pick up on some of the speaker's tendencies (e.g., using the onomatopoeia "Erm").

# F   Cross-lingual Augmentation Experimental Setup

The original FBTOD dataset contains an English dataset with 30,521 training instances, 4,181 evaluation instances, and 8,621 testing instances. It also contains a Spanish dataset with 3,617 training instances, 1,983 evaluation instances, and 3,043 testing instances. For both languages, we created versions of the training set that had $1\%$, $5\%$, and $10\%$ of the original number of training instances. For each percentage, we randomly sampled that proportion of training examples for each intent label. In cases where that proportion would result in a number smaller than $1.0$, we ensured that there would be at least one training instance.

For augmentation, we use the prompt given in Figure 11. We perform beam search, taking up to three sequences as augmented data and rejecting any duplicate examples. For classification, we fine-tune XLMRoBERTa with a maximum sequence length of 128, 80 training epochs, and an initial learning rate of 5e-5.

# G   Entropy-based Weak Filtering

We hypothesized that two categories of synthetic data could improve to eventual task performance. The first is simply *correct* data — synthetic training instances which match the intended instance label. However, we determine this correctness using an imperfect classifier. Thus, the second category is *"hard to learn"* data. This follows from early work [Guo and Viktor, 2004] finding that classification performance can be improved by focusing on augmenting datasets with "hard to learn" examples. However Guo and Viktor [2004] identify such difficult examples post hoc through classifier performance. Solely evaluating classifier performance for each example is prohibitively expensive for large models on large datasets. Thus, in our case, we make our decisions based on uncertainty. Out of those instances that the classifier states are incorrect, we do not filter out the ones for which the classification is made with high uncertainty, hypothesizing that uncertainty is a strong proxy for data points' learning difficulty, which may be more useful in training the next iteration of the classifier.

There are several ways to consider how to quantify uncertainty. Ott et al. [2018] quantified uncertainty for neural machine translation post hoc by comparing generation methods' sequence-level probability mass coverage of ground-truth translations. While these methods may be appropriate for translation or even response generation-related tasks, there is no reference point for what the ideal augmented data is.

Other approaches have looked specifically at quantifying dataset uncertainty using Bayesian Neural Networks [Xiao and Wang, 2019, Chen et al., 2020]. However, dataset level approaches are not necessarily appropriate for augmentation, where one needs to make inferences about data quality at the instance level. Moreover, approaches requiring Bayesian Neural Networks do not achieve state-of-the-art performance on the DAILYDIALOG classification tasks.

On the other hand, entropy maximization has long been used as an information theoretic technique for maximizing uncertainty [Seidenfeld, 1986, Wang, 2008]. Even in natural language processing, Csáky et al. [2019] performs data filtering using entropy to improve diversity by specifically removing generic data points. They achieve promising results in dialogue generation, and moreover, entropy computation is directly applicable at the instance level. Therefore, we adopt entropy into our framework to identify uncertain and difficult data points.

