# OpenReview forum: "Weakly Supervised Data Augmentation Through Prompting for Dialogue Understanding"
_NeurIPS.cc/2022/Workshop/SyntheticData4ML — Neurips 2022 SyntheticData4ML_

### Official Review · Reviewer_JMCz · 2022-10-12
**Interesting approach that demonstrates promise**

**Rating:** 8
**Confidence:** 3

**Review:**

Interesting idea to use a classifier post inference in the feedback loop. I found it to be creative, and apparently easy to use as a plug-in method. The consistent improvement across accuracy and F1 scores for the Spanish dataset is also attractive.

---

### Official Review · Reviewer_91pU · 2022-10-17
**Interesting problem to tackle, but doubtful about the motivation of the proposed approach**

**Rating:** 5
**Confidence:** 4

**Review:**

Summary
This paper proposes a filtering technique to select the best quality synthetic data generated by prompts for the dialogue task. They evaluate various augmentation methods on the classification tasks in DAILYDIALOG and the intent detection task of FBTOD.
What the paper addresses an existing problem in dialogue augmentation, however whether it is the correct approach or not is entirely clear to me (see question 1).

##########################################################################

Positive points about this paper

Addressing the problem of out-of-distribution data generation is an interesting idea. Augmentation is a valid approach to address this lack of data in the dialogue task.
The paper covers the literature well and the experiments showcase the usability of the approach. It is well-written and experiments are clearly defined for the most part.

##########################################################################

Concerns and questions:

- My main concern for this paper is as follows: The authors say that prompted augmentations are less potent because they lack task-knowledge, which can be distilled from ground-truth (“gold”) samples by training an auxiliary model. They then use that model to filter out inconsistent generated utterances. The filtering model would work with the task knowledge ONLY if there is already relevant and in-distribution data generated by the LM. It doesn't necessarily address the problem of lack of task-knowledge in the augmentation process. Isn't that the main problem this paper is trying to address?

- What is the impact of the prompt you used on the results? How much effort went into engineering the prompt? Did you try different variations or did you not "finetune" the prompt?

---

### Official Review · Reviewer_p4X5 · 2022-10-18
**Few-shot data augmentation for dialogue understanding**

**Rating:** 7
**Confidence:** 4

**Review:**

In this work, the authors present a few-shot data augmentation framework for dialogue understanding by prompting large pre-trained language models. They further introduce a weakly supervised labeler to filter prospective data points. Models fine-tuned on their augmented data approach or surpass existing state-of-the-art performance on DAILYDIALOG and FACEBOOK MULTILINGUAL TASK-ORIENTED
DIALOGUE datasets.

---

### Meta-Review · Area_Chair_Ytcf · 2022-10-20

**Recommendation:** Accept

**Review:**

The reviewers agree that the paper proposes an interesting idea and is well-written, hence I recommend to accept this paper. If accepted, I would urge the authors to address the two concerns of reviewer 91pU in the camera-ready version.